# Imaging Findings in Patients with Immune Checkpoint Inhibitor-Induced Arthritis

**DOI:** 10.3390/diagnostics12081961

**Published:** 2022-08-13

**Authors:** Andrés Ponce, Beatriz Frade-Sosa, Juan C. Sarmiento-Monroy, Nuria Sapena, Julio Ramírez, Ana Belén Azuaga, Rosa Morlà, Virginia Ruiz-Esquide, Juan D. Cañete, Raimon Sanmartí, José A. Gómez-Puerta

**Affiliations:** Department of Rheumatology, Hospital Clinic of Barcelona, 08036 Barcelona, Spain

**Keywords:** immunotherapy, immune-related adverse events (irAE), cancer treatment, diagnosis, immune checkpoint inhibitors, immune oncology, toxicity

## Abstract

Immune checkpoint inhibitor (ICI)-induced arthritis is an increasingly recognized adverse event in patients with oncologic disease during immunotherapy. Four patterns are well described, including rheumatoid arthritis (RA)-like, polymyalgia rheumatica (PMR)-like, psoriatic arthritis (PsA)-like, and oligo-monoarthritis, among others. Despite better clinical recognition of these syndromes, information about the main imaging findings is limited. Methods: We conducted a retrospective observational study including all adult patients referred to the Rheumatology Department of a single-center due to ICI-induced arthritis who underwent imaging studies [ultrasound (US), magnetic resonance imaging (MRI), and ^18^F-FDG PET/CT)] between January 2017 and January 2022. Results: Nineteen patients with ICI-induced arthritis with at least one diagnostic imaging assessment were identified (15 US, 4 MRI, 2 ^18^F-FDG PET/CT). Most patients were male (84.2%), with a median age at inclusion of 73 years. The main underlying diagnoses for ICI treatment were melanoma in five cases. The distribution of ICI-induced arthritis was as follows: PMR-like (5, 26.2%), RA-like (4, 21.1%), PsA-like (4, 21.1%), and others (6, 31.6%). All RA-like patients had US findings indistinguishable from conventional RA patients. In addition, 3/5 (60%) of PMR-like patients had significant involvement of the hands and wrists. Abnormal findings on MRI or PET-CT were reported by clinical symptoms. No erosions or myofascitis were seen. Conclusions: ICI-induced arthritis patients present inflammatory patterns on imaging studies similar to conventional inflammatory arthropathies, and therefore these syndromes should be followed carefully and treated according to these findings.

## 1. Introduction

Oncologic treatment with immune checkpoint inhibitors (ICI) includes cytotoxic T-lymphocyte-associated antigen-4 (anti-CTLA-4) and the programmed death-1 (PD-1)/programmed death-ligand 1 (PD-L1) axis, among others. Their appearance has meant a paradigmatic change in oncology treatment, producing significant survival benefits in patients with different types of cancer [1].

Considering their mechanism of action, it is not surprising that ICI therapy might induce several side effects, such as immune-related adverse events (irAEs). Almost every organ may be affected, with the severity ranging from mild to life-threatening [2]. Musculo-skeletal irAEs are not uncommon and are increasingly recognized, including ICI-induced arthritis [3,4,5,6], of which arthralgia is the most common, with a prevalence ranging from 5–20% [7], but other rheumatic syndromes have been reported [8,9], including various kinds of inflammatory arthritis (polyarthritis resembling rheumatoid arthritis (RA), reactive arthritis-like disease or large joint oligoarthritis with or without axial disease), a clinical syndrome resembling polymyalgia rheumatica (PMR-like), myositis [8,9], vasculitis [10] or sarcoidosis-like [11]. Less commonly, some cases of systemic lupus erythematosus [12], fasciitis [13], systemic sclerosis, antiphospholipid syndrome [14] and dermatomyositis [13] have been reported among others.

The clinical course in ICI-induced arthritis is variable. Recently, our group described three different patterns [15]. More than half the patients had a self-limited flare, around one quarter had an intermittent course with flares and remission, and around 20% had a chronic persistent pattern, as occurs in inflammatory arthritis. Some patients had a chronic course after ICI cessation [16]. 

In recent years, several scientific societies have developed general recommendations for the treatment and follow-up of these patients [17,18]. However, there is a lack of knowledge about the role that imaging tests can play in this disease.

Ultrasonography (US) and magnetic resonance imaging (MRI) have become important modalities for the diagnosis and monitoring of inflammatory arthritis, as they are able to identify inflammatory features (such as synovitis, effusions, bursitis, tendonitis, tendosynovitis), enthesopathy, and structural damages (such as erosions) [19]. MRI is also a very good imaging technique for determining muscular involvement in patients with inflammatory myopathies [20]. Positron emission tomography with 2-deoxy-2-[fluorine-18] fluoro-D-glucose integrated with computed tomography (^18^F-FDG PET/CT) is an imaging modality that is well-established in oncology and has an expanding role in the assessment of inflammatory conditions in rheumatology [21]. ^18^F-FDG PET/CT has high sensitivity in detecting rheumatic conditions, both articular and systemic, and in determining their extent and severity. 

There are few reports about the role that imaging tests play in patients who develop irAEs [22,23,24,25,26,27,28]. This study aimed to describe the imaging test findings associated with the different clinical patterns in patients with ICI-induced arthritis in patients treated at our hospital. We also reviewed the literature about different imaging assessments and the characteristics of ICI-induced arthritis.

## 2. Materials and Methods

We conducted a retrospective observational study including all adult patients referred to the Rheumatology Department of our center due to the onset of rheumatic syndromes related to ICI treatment who underwent imaging studies (US, MRI and/or ^18^F-FDG PET/CT) between January 2017 and January 2022. 

Demographic data, history of rheumatic diseases, ICI indication and type, presence of non-rheumatic irAEs, disease manifestations at irAE onset, and treatment were collected.

First, we made a descriptive analysis according to the findings of the US study. Sonographic assessments were made using high-sensitivity US equipment (MyLab9^®^; Esaote, Genoa, Italy), with a longitudinal probe, frequency range from 5–14 MHz and a pulse repetition frequency between 800 and 900 Hz, by a single experienced sonographer (AP). Joint musculoskeletal US findings were defined according to published Outcome Measures in Rheumatoid Arthritis Clinical Trials (OMERACT) definitions [29]. The US examination evaluated symptomatic joints for synovial hypertrophy (SH) and intra-articular power Doppler (PD) signaling according to EULAR guidelines [19]. SH and PD signals were graded using a four-grade semi-quantitative scoring system (0 = no, 1 = mild, 2 = moderate and 3 = severe) according to the methodology of Szkudlarek et al. [30]. The highest SH and PD grade detected during the scans was adopted as representative of each joint, respectively. 

We classified clinical syndromes according to four categories: (a) polyarthritis- RA-like, (b) PMR-like, (c) psoriatic arthritis-like (PsA-like) in patients with oligo/polyarthritis with or without enthesitis and tenosynovitis, (d) other types of arthritis (oligoarthritis, monoarthritis or only tenosynovitis).

We reviewed MRI and ^18^F-FDG PET/CT assessments made by the rheumatologists of all patients included and described the imaging findings. 

All patients gave written informed consent. The study was approved by the Hospital Clinic Institutional Review Board (HCB/2021/0901). 

We also reviewed the literature on the different imaging techniques and rheumatological irAEs. The bibliographic search was conducted in the MEDLINE database up to June 2022. Publications were identified using the following MeSH terms: “Immune checkpoint inhibitors” and [“arthritis” or “arthralgia” or “joint”] and “ultrasonography”; “Immune checkpoint inhibitors” and [“arthritis” or “arthralgia” or “joint”] and “magnetic resonance imaging”; and “Immune checkpoint inhibitors” and [“arthritis” or “arthralgia” or “joint”] and “Positron Emission Tomography Computed Tomography”.

## 3. Results

Nineteen patients were included (15 US, 4 MRI, and 2 ^18^F-FDG PET/CT). Most patients were male (84.2%) and the median age at inclusion was 73 (67–76) years. The main underlying diagnoses for ICI treatment were melanoma in five cases, lung carcinoma, urothelial cancer, and liver carcinoma in three cases each. Most patients received ICI treatment as monotherapy, with Pembrolizumab (6, 31.6%), Nivolumab (3, 15.8%), and Anti- T-cell immunoglobulin and mucin domain 3 (Anti-TIM3) (3, 15.8%) being the most frequently prescribed treatments. Four patients (21.1%) received combined treatment.

The distribution of ICI-induced arthritis was as follows: PMR-like (5, 26.2%), RA-like (4, 21.1%), psoriatic arthritis (PsA-like) (4, 21.1%), and others (6, 31.6%). Arthritis had a self-limited course in 47.4% of patients. The mean time between ICI initiation and rheumatic irAE was four (2–10) months. Regarding the treatment of rheumatological syndromes, nine (47.4%) patients received glucocorticoids (Prednisone mean dose 17 mg/day) and four conventional synthetic DMARDs (two Methotrexate, one Sulfasalazine, one Hydroxychloroquine). In five patients, ICI were halted due to the rheumatic irAE. Of patients with RA-like patterns, none were seropositive for rheumatoid factor (anti-RF) or anti-citrullinated protein antibodies (anti-CCP).

Six patients had a rheumatologic/inflammatory disease before ICI, including chondrocalcinosis (2 cases), HLA-B27 associated uveitis (1 case), psoriasis (1 case), fibromyalgia (1 case) and osteoarthritis (1 case). Five patients presented non-rheumatic irAEs before the appearance of rheumatic syndrome, including sarcoidosis (1 case), colitis (1), Sweet’s syndrome (1), sensitive axonal polyneuropathy (1), and hypothyroidism (1) (Table 1).

### 3.1. Patients with US Assessment

US was performed in 15 patients, with different clinical patterns: five PMR-like, four RA-like, two PsA-like, two oligoarthritis and two monoarthritis (Table 2 and Table 3).

The US findings in RA-like patients were synovitis and tenosynovitis of the wrists and flexor tendon of the fingers. One patient also had synovitis in the knees and ankles, without involvement of the feet. Although they were seronegative (RF and anti-CCP), patients presented significant SH and PD signals, except for one patient treated with high doses of prednisone (40 mg/d). A set of US images with the most representative changes are shown in (Figure 1). 

Five patients had a PMR-like syndrome. In three, the imaging test was made before the start of glucocorticoid treatment. US showed synovitis of the hip and trochanteric bursitis, and bilateral inflammatory bicep tenosynovitis, in one patient. In the other two patients, the shoulders and hip joints were studied by PET/CT scan, showing a periarticular pattern in the shoulder and pelvic girdle (see further comments below). We also identified sonographic changes in the wrists and hands in patients with PMR-like disease, affecting not only the wrists but also the MCPs, and finger tenosynovitis (Figure 2).

One patient had a previous diagnosis of seronegative arthritis, associated with knee arthritis. A synovial fluid analysis showed the presence of pyrophosphate crystals. 

One patient had dactylitis in two fingers. US revealed soft tissue thickening with PD signal, subcutaneous edema, and inflammatory tenosynovitis in the flexors of the hand (Figure 3). The two patients with monoarthritis (carpus and knee) had a history of rheumatic disease (chondrocalcinosis and HLA-B27 associated uveitis). In these patients, SH and grade 2 PD signals were detected. The distribution of joint involvement (synovitis and tenosynovitis) according to the arthritis patterns after US evaluation is shown in (Figure 4).

### 3.2. Patients with MRI Assessment

Four patients underwent MRI assessment. Two patients received treatment with Anti-TIM3, one with Pembrolizumab, and one with Durvalumab. In these patients, ICI-induced arthritis appeared after a mean time of 15.5 months. Abnormal findings were reported in three out of four patients, all in accordance with clinical symptoms. One patient presented with a PsA-like pattern and had signs of distal bone edema in the fingers (Figure 5). No erosions or myofascitis were reported. The general characteristics of previous cases reported with rheumatic irAEs and MRI assessment are shown in Appendix A. 

### 3.3. Patients with PET/CT Scan Assessment

Two male patients had PET/CT scans to study rheumatic irAE secondary to the use of ICI. Patient 1 had melanoma and was treated with Pembrolizumab. Patient 2 had prostate carcinoma and was treated with Atezolizumab and Cabozantinib. Neither had previous rheumatic disease. Both patients presented clinical PMR. PET-CT scans showed symmetric FDG uptake in the shoulder, hip joints, and greater trochanter and ischial tuberosities. One also showed FDG uptake in the left knee (Figure 6). The previous and our reports of PET/CT scan findings in rheumatic irAEs are shown in Appendix A. 

## 4. Discussion

Few studies have focused on imaging findings in patients with irAEs induced by ICI. We describe several US, MRI, and ^18^F-FDG PET/CT findings in patients with ICI-induced arthritis and their association with the clinical pattern of presentation.

US aids a better definition of the spectrum of clinical findings on physical examination or helps identify inflammation in paucisymptomatic patients, including synovitis, tenosynovitis, bursitis, and enthesopathy. So far, only a few studies have reported US findings in ICI-induced arthritis, with heterogeneous results [26,27,28].

In our case series, US was performed after clinical assessment and as complementary information after physical examination. The US findings in patients with RA-like syndrome consisted of marked synovitis/tenosynovitis, although they were seronegative (anti-RF and ant-CCP), and presented high grades of SH and PD signals, with sonographic findings indistinguishable from typical RA. These US findings do not differ from those found in other series of patients with synovitis/tenosynovitis induced by ICI and studied by US [23].

It has been reported that ICI-induced PMR has US and FDG-PET/CT results comparable to those seen in regular PMR [26]. We also found characteristic US findings in PMR-like ICI-induced patients similar to those previously described (i.e., glenohumeral and hip joint synovitis, bicipital tenosynovitis, and subacromial bursitis) [26]. Peripheral symmetrical synovitis and tenosynovitis, clinically subtle or not evident at all, in addition to rhizomelic involvement was a notable finding. Although peripheral synovitis and tenosynovitis are frequently reported in PMR, it is usually oligoarticular and transient [31]. As in other studies of typical PMR [32], we found that in patients with ICI-induced PMR, tenosynovitis of the flexor tendons of the hand was the most frequent US finding of peripheral involvement. Interestingly, in treated patients, we found not synovitis, tenosynovitis, or bursitis in the shoulders and pelvic girdles, which seemed to have resolved with glucocorticoid treatment. However, peripheral synovitis/tenosynovitis in the hand joints remained evident.

In particular, in one patient with dactylitis (Figure 3), US findings included soft tissue thickening, subcutaneous edema, flexor tenosynovitis, and synovitis, without major differences with previously-described typical PsA [33]. US findings of oligoarthritis and monoarthritis patterns showed high levels of SH and PD signals, with no particularities with respect to inflammatory arthritis.

MRI has demonstrated a superior sensitivity for assessing inflammation and structural damage, compared with the clinical examination, X-rays, and US in patients with chronic inflammatory arthropathies such as RA, PsA, and spondyloarthropathies, among others. However, it has limitations to access, including the waiting time from the patient’s visit to the diagnostic test and its cost, among others.

The use of MRI in RA is well standardized. For instance, the OMERACT group established a scoring system (RAMRIS) for patients with RA, defining a core set of basic MRI sequences and MRI definitions including synovitis, bone erosions, osteitis, joint space narrowing, and tenosynovitis.

MRI definitions and findings are not well standardized in ICI-induced arthritis. To the best of our knowledge, around 20 cases of ICI-induced arthritis with MRI assessments have been reported [16,24,34,35,36]. 

Subedl et al. [34] reported MRI findings in eight patients with ICI-induced arthritis at NIH (Bethesda, USA). The cohort included a heterogeneous group of patients with MRI performed for a variety of clinical symptoms. Joint symptoms started a median of 10 weeks after ICI therapy. Erosions were reported in three patients. However, one patient was positive for anti-CCP antibodies without a history of RA and another patient had a history of RA in remission before CPI treatment. The remaining findings included synovitis, tenosynovitis, and synovial thickening in the joints involved (mainly the hands, wrists, knees, and ankles).

Daoussis et al. [24] identified 10 out of 130 patients who developed musculoskeletal irAEs related to ICI. The median time from treatment initiation to irAE was 2.5 months. Most patients complained of polyarthritis predominantly affecting the hands, wrists, and feet. Eight patients underwent MRI assessments. The authors reported three different MRI patterns: (1) prominent joint involvement (2) prominent periarticular involvement and (3) myofascitis. MRI identified signs of fasciitis of the surrounding tissues of the joints involved, even in patients who presented with arthritis, suggesting a differential MRI pattern in ICI-induced arthritis. In our series, no case with MRI assessment had signs of myofascitis. 

Some other anecdotal cases with MRI abnormalities have been reported affecting the axial structures. More details are shown in Appendix A. In our patients, US was the initial imaging test for evaluating the inflammatory joint disease in ICI-induced arthritis, due to easy accessibility. In our experience, US resolved clinical discrepancies in most cases. We propose that US could be used as the initial exploration, reserving MRI for unresolved issues, suspected early erosive disease, or muscle or myofascial involvement [24,34]. 

^18^F-FDG PET/CT is frequently used to evaluate tumors and treatment efficacy. Although there are no guidelines or consensus for the detection of irAEs by imaging, it is also a useful technique to evaluate irAEs [25]. ^18^F-FDG PET has high sensitivity in detecting rheumatic irAEs and determining their extent and severity, information that is important in patient management [25]. However, the incidence of synovitis with ^18^F-FDG PET/CT could be under-estimated, because not all joint regions, including the feet, are covered by this technique [37].

^18^F-FDG PET-positive rheumatic irAEs, such as arthritis, myositis, tenosynovitis, and PMR have been reported [25,26,27,28,37,38,39,40,41]. The main characteristics and imaging findings are summarized in Appendix A. These changes do not only appear in clinically symptomatic patients [25,27]. The meaning of this hypermetabolic uptake is unclear. 

Interestingly, some patients who underwent ^18^F-FDG PET before ICI therapy had some articular irAE-specific changes, which were exacerbated after immunotherapy [25,26,27,40]. As suggested by van der Geest et al., we agree that although this mild metabolic activity may also be seen in non-inflammatory conditions, it could suggest that low-grade, subclinical inflammation may already have been present at these sites before ICI therapy and was potentiated after treatment [26]. Inflammatory changes in multiple joints found by PET/CT in patients in this study probably reflected an autoimmune cellular attack on the synovia [28]. Here we report two consecutive patients with ICI-induced arthritis who presented PMR-like disease with ^18^F-FDG PET/CT showing symmetric FDG uptake in the shoulder, hip joints, greater trochanter, and ischial tuberosities, in accordance with the clinical signs. 

^18^F-FDG PET has also high sensitivity in determining the resolution of inflammation, which is essential in the management of patients starting glucocorticoid treatment and interrupting or withdrawing ICI treatment. It has been found that, after discontinuation of immunotherapy and treatment with glucocorticoids, the changes found on PET were resolved [25,27,28]. Early detection of irAEs allows early management. Physicians should be aware of these irAEs when interpreting ^18^F-FDG PET in patients undergoing ICI therapy. Careful analysis of ^18^F-FDG PET-CT, focusing on the detection of signs of rheumatic irAEs could potentially help oncologists during follow-up who should request a prompt assessment by rheumatologists. 

^18^F-FDG PET detectable irAEs can also provide additional information of clinical value. It has been suggested that PET/CT-detectable irAEs, including rheumatic irAEs, could provide an early indicator of the efficacy of immunotherapy and better oncologic outcomes [38,42], although validation is required in larger cohorts with a longer follow-up.

This study has some limitations. We only conducted imaging tests in symptomatic patients. As mentioned above, sensitive techniques such as MRI or PET/CT scan can detect articular and periarticular subclinical inflammation in patients under ICI therapy, and therefore articular symptoms could be underestimated in the physical examination.

We included a relatively small cohort of patients with ICI-induced arthritis. The number of studies carried out, especially MRI and 18F-FDG PET/CT, allow a description of the most representative findings in patients with ICI-induced arthritis. A study of a larger number of patients to characterize these imaging findings better is required. However, so far, our sample is one of the largest series of patients with irAEs with imaging assessments. Given the nature of our study, we did not follow a pre-established protocol for the treatment of ICI-induced arthritis, and therefore many patients were analyzed under glucocorticoid treatment. We assumed that we would find more accentuated changes in patients without background therapy. Finally, since there is no well-defined US pattern for ICI-induced arthritis, we classified sonographic findings according to four previously described clinical patterns. Given the heterogeneity in the different domains and terms used to clinically characterize ICI-induced arthritis, an OMERACT working group was created to homogenize the different patterns [43]. Due to a lack of information in the imaging field, a further phase incorporating a core setting with the main imaging findings would be desirable.

## 5. Conclusions

Imaging techniques are helpful monitoring tools for diagnosing and following patients with ICI-induced arthritis. Since many patients have mild clinical symptoms or are under glucocorticoid treatment, a better definition of articular and periarticular structures is extremely useful in patient assessments. US findings in our study disclosed significant inflammatory changes, for instance, in patients with RA-like and PMR-like disease, as occur in chronic inflammatory arthropathies. These findings have implications for the diagnosis, treatment and prognosis, requiring, in some cases, DMARD therapies. Careful analysis of ^18^F-FDG PET, focusing on detecting signs of rheumatic irAEs could potentially help oncologists during follow-up to request a prompt assessment by rheumatologists. We propose that the US could be used as the initial examination and reserve MRI for unresolved issues, suspected early erosive disease, or muscle or myofascial involvement. The wider of more sensitive techniques, such as MRI and PET/CT scan, will allow for the better recognition and definition of these novel and challenging rheumatic syndromes.

## Figures and Tables

**Figure 1 diagnostics-12-01961-f001:**
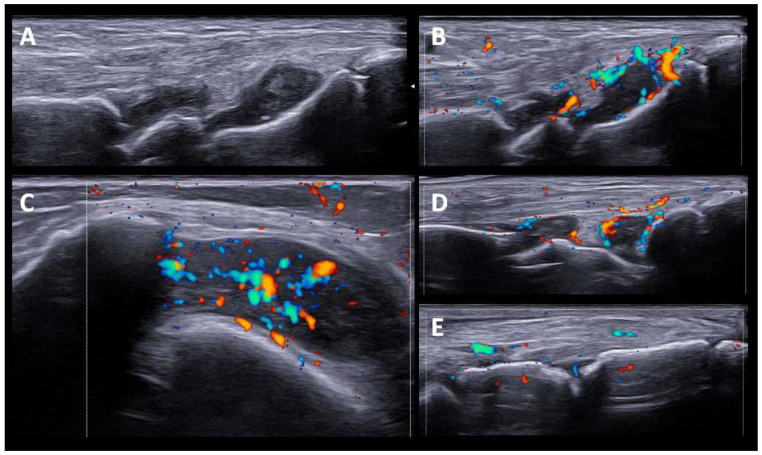
**US findings in RA-like pattern.** (**A**,**B**). Carpal joint with synovial hypertrophy grade 2 and power doppler signal grade 2. (**C**). Knee suprapatellar recess shows synovial hypertrophy grade 3, and Power Doppler signal grade 2. (**D**,**E**). Tibiotalar, subtalar and tarsal joints with moderate synovitis and Power Doppler signal grade 2.

**Figure 2 diagnostics-12-01961-f002:**
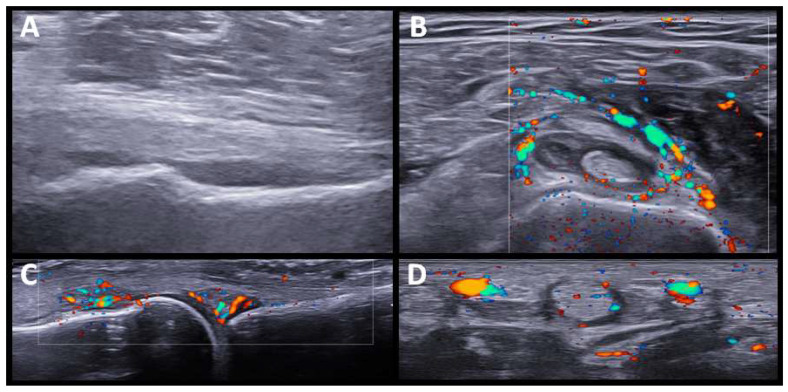
**US findings in PMR-like pattern.** (**A**). Hip joint (longitudinal view) with moderate effusion. (**B**). Bicipital tenosynovitis with power doppler signal grade 2. (**C**). Synovitis at second MCP joint with Power Doppler grade 2. (**D**). Tenosynovitis of hand flexor tendon with Power Doppler grade 2.

**Figure 3 diagnostics-12-01961-f003:**
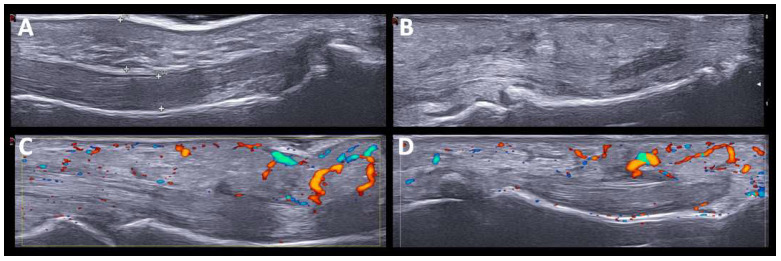
**US findings in PsA-like pattern.** Hand dactylitis. The US findings in PsA-like pattern. Hand dactylitis. The US findings include soft tissue thickening, subcutaneous edema and flexor tenosynovitis (**A**,**B**) with Power Doppler signal (**C**,**D**).

**Figure 4 diagnostics-12-01961-f004:**
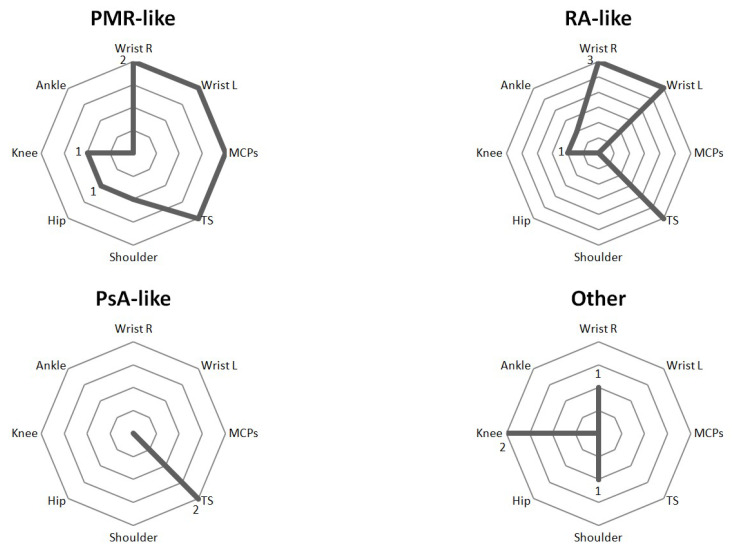
**Spider diagram of US findings according to ICI-induced arthritis patterns.** MCP: metacarpophalangeal joint. PMR-Like: Polymyalgia rheumatica like, PsA-like: psoriatic arthritis like, RA-like: rheumatoid arthritis like, TS: tenosynovitis.

**Figure 5 diagnostics-12-01961-f005:**
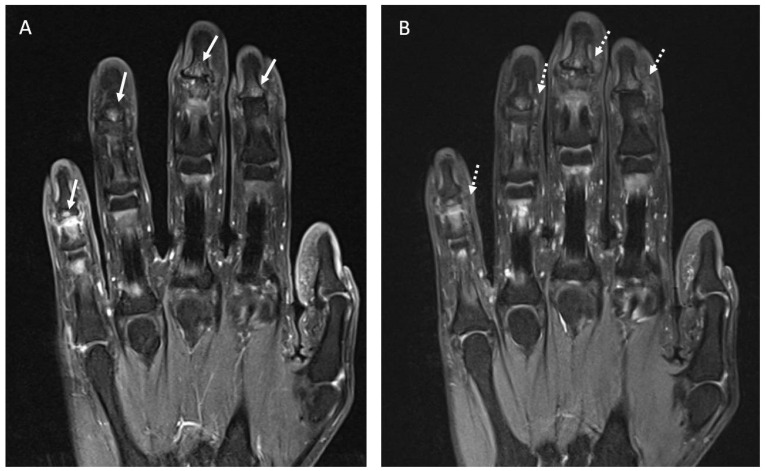
(**A**). Coronal fat suppressed T1-weighted image of the hand following IV contrast administration shows small areas of contrast enhancement of the subcortical articular margin (arrows) of the distal interphalangeal joints in the second, third, fourth and fifth fingers consistent with inflammatory activity. (**B**). Coronal fat suppressed DP-weighted image of the hand shows areas of bone edema of the subcortical articular margin (dashed arrows) of the distal interphalangeal joints in the second, third, fourth and fifth fingers.

**Figure 6 diagnostics-12-01961-f006:**
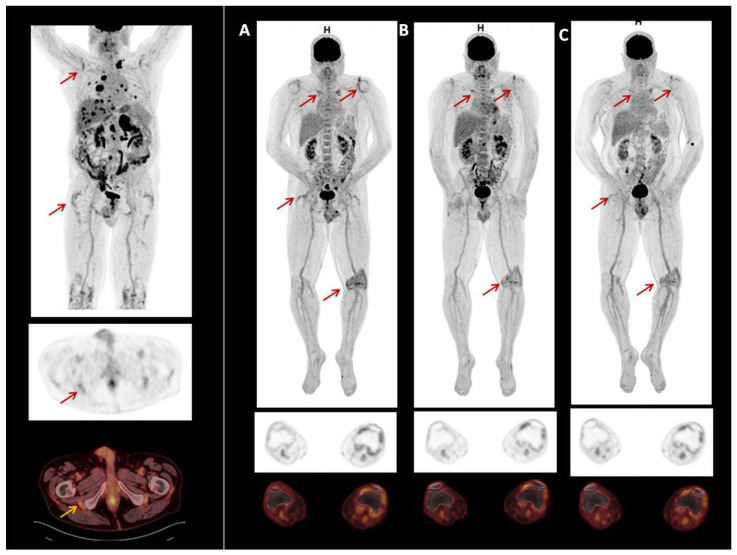
^18^F-FDG PET/CT image of patient with prostate carcinoma treated with Atezolizumab and Cabozatinib. Symmetric FDG uptake shoulder, hip joints, greater trochanter and ischial tuberosities. Also show, FDG uptake in left knee (**A**). In 2020, due to irAEs ICI were discontinued. After discontinuation of treatment, uptake decreased (**B**,**C**).

**Table 1 diagnostics-12-01961-t001:** General characteristics, type of cancer and ICI molecules.

Characteristic	n/19 (%) (Median IQR)
Gender (Male)	16 (84.2)
Age, years	73 (67–76)
Mean time from ICI initiation and irAE onset, months	4 (2–10)
Type of cancer	
Melanoma	5 (26.3)
Lung	3 (15.8)
Liver	3 (15.8)
Urothelial	3 (15.8)
Prostate	2 (10.5)
Acute leukemia	2 (10.5)
Myelodysplastic syndrome	1 (5.3)
Previous rheumatic diseases	
Chondrocalcinosis	2 (10.5)
HLA-B27 associated uveitis	1 (5.3)
Psoriasis	1 (5.3)
Fibromyalgia	1 (5.3)
Osteoarthritis	1 (5.3)
Type of Checkpoint inhibitors	
Monotherapy	
Pembrolizumab	6 (31.6)
Anti TIM3	3 (15.8)
Nivolumab	3 (15.8)
Atezolizumab	2 (10.5)
Durvalumab	1 (5.3)
Combined therapy	
Pembrolizumab + Epacadostat	1 (5.3)
Atezolizumab + Cabozantinib	1 (5.3)
Nivolumab + Rigorafenib	1 (5.3)
Ipilimumab + Nivolumab	1 (5.3)

Anti-TIM3: Anti T cell immunoglobulin mucin domain 3; IQR: Interquartile Range.

**Table 2 diagnostics-12-01961-t002:** Clinical and demographic characteristics in patients with US assessment.

Case	Age	Sex	Cancer	Treatment	Prednisone Dose (mg/d)	Clinical Pattern	Pre-Rheum	T from CPI and irAEs (Months)	Previous irAEs
1	78	M	Melanoma	Pembrolizumab	-	PMR-like	-	1	Colitis
2	74	M	Urinary tract	Nivolumab + Ipilumab	-	RA-like	-	1	-
3	78	M	Prostate	Nivolumab	10	RA-like	-	12	-
4	75	F	Melanoma	Nivolumab	60	RA-like	-	2	-
5	72	M	Lung	Nivolumab	40	RA-like	-	2	-
6	77	M	Vesical	Pembrolizumab	5	PMR-like	-	2	-
7	60	M	Melanoma	Nivolumab	-	PMR-like	Seronegative arthrtitis	6	-
8	73	M	Prostate	Atezolizumab	20	PMR-like	-	1	-
9	76	M	Hepatocarcinoma	Atezolizumab	-	PMR-like	-	1	-
10	53	F	Melanoma	Pembrolizumab	-	PsA-like	-	31	-
11	68	F	Melanoma	Pembrolizumab	7,5	Oligoarthritis	Fibromyalgia	11	-
12	58	M	Hepatocarcinoma	Durvalumab	10	Oligoarthritis	-	20	Peripheral polyneuropathy
13	74	M	Lung	Atezolizumab	-	Monoarthritis	Chondrocalcinosis	5	-
14	72	M	Leukemia	Anti-TIM3	-	Monoarthritis	Uveitis HLA-B27+	9	-
15	59	M	Hepatocarcinoma	Nivolumab	-	PsA-Like (Dactylitis)	-	7	-

Pre-Rheum: previous rheumatic diseases. T. from CPI and irAEs: time from checkpoint inhibitor and immune-related adverse events: Anti-TIM3: Anti T cell immunoglobulin mucin domain 3, PMR-Like: Polymyalgia rheumatica like, RA-like: Rheumatoid Arthritis like, PsA-like: Psoriatic arthritis-like.

**Table 3 diagnostics-12-01961-t003:** US findings according to anatomical regions.

Case	Clinical Pattern	R.Wrist Grade SH	R.Wrist Grade PD	L.Wrist Grade SH	L.Wrist Grade PD	MCP Grade SH	MCP Grade PD	TS Flexor Hands (Number)	Shoulder Syndrome	Hip	Knee	Ankle	Feet	Others
1	PMR-like	2	3	2	3	2	3	0	no	no	no	no	no	
2	RA-like	3	2	3	2	0	0	0	no	no	no	no	no	
3	RA-like	2	1	2	1	0	0	8	no	no	yes	yes	no	
4	RA-like	3	2	3	2	0	0	1	no	no	no	no	no	
5	RA-like	0	0	0	0	0	0	6	no	no	no	no	no	
6	PMR-like	2	0	2	1	0	0	6	no	no	no	no	no	
7	PMR-like	0	0	0	0	0	0	0	no	no	yes	no	no	Calcium pyrophosphate crystals PET/CT *
8	PMR-like	0	0	0	0	2	2	8	no	no	no	no	no	PET/CT *
9	PMR-like	0	0	0	0	0	0	0	yes	yes	no	no	no	Trochanteric bursitis
10	PsA/Oligo	0	0	0	0	0	0	0	no	no	no	no	no	
11	PsA/Oligo	0	0	0	0	0	0	1	no	no	no	no	no	
12	Oligo arthritis	0	0	0	0	0	0	0	yes	no	yes	no	no	
13	Monoarthritis	2	2	0	0	0	0	0	no	no	no	no	no	
14	Monoarthritis	0	0	0	0	0	0	0	no	no	yes	no	no	
15	PsA/Oligo	0	0	0	0	0	0	2	no	no	no	no	no	US Dactylitis

* Periarticular pattern in shoulders and hip joints; SH: Synovial Hypertrophy; MCP: metacarpophalangeal joint, PD: Power Doppler signal. PMR-Like: Polymyalgia rheumatica like, PsA-like: Psoriatic arthritis like, RA-like: Rheumatoid arthritis like, TS: tenosynovitis. SH and PD signals were graded using a four-grade semi-quantitative scoring system (0 = no, 1 = mild, 2 = moderate and 3 = severe) according to the methodology of Szkudlarek et al. [30]. The highest SH and PD grade detected during the scans was adopted as representative of each joint.

## Data Availability

Not applicable.

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
