# Peer review of "Imaging Findings in Patients with Immune Checkpoint Inhibitor-Induced Arthritis"

_diagnostics, 2022, doi:10.3390/diagnostics12081961_

Round 1

Reviewer 1 Report

Very nice topic and very good 'pictorial' paper.

I suggest You:

- Please revise all the key-words according to Mesh terms. Immune oncology and immune-related adverse events do not seem to be Mesh terms.

- Please add in the abstract more numerical/statistical data on your research even if this is focused on few patients: e.g. RA-like patients had US findings indistinguishable from conventional RA patients (how many%). Also, PMR-like patients had important involvement in the hands and wrists (how many%).

- Since only a few patients have MRI and PET I would not state that you were able to depict the main findings of these patients with these tools (discussion).

- Do you think that US is enough for the evaluation of these inflammatory joint disease? When MRI is needed? Please add a brief discussion on this with your experience, and citing also literature data.

Author Response

Very nice topic and very good 'pictorial' paper.

We thank the reviewer for the feedback.

I suggest You:

  1. Please revise all the key-words according to Mesh terms. Immune oncology and immune-related adverse events do not seem to be Mesh terms.

       Thanks for the feedback. We have reviewed all keywords used in the manuscript. We have used the terms suggested by the editors for this special issue on checkpoint inhibitors. Unfortunately, there are no MeSH terms for "Immune oncology" and "immune-related adverse events".

  1. Please add in the abstract more numerical/statistical data on your research even if this is focused on few patients: e.g. RA-like patients had US findings indistinguishable from conventional RA patients (how many%). Also, PMR-like patients had important involvement in the hands and wrists (how many%).

                According to the suggestion, we have added in the abstract more statistical data on our research.

We now state “Results: Nineteen patients with ICI-induced arthritis with at least one diagnostic imaging assessment were identified (15 US, 4 MRI, 2 18F-FDG PET/CT). Most of the patients were male (84.2%), with a median age at the moment of inclusion of 73 years. The main underlying diagnoses for ICI treatment were melanoma in 5 cases. The distribution of ICI-induced arthritis was as follows: PMR-like (5, 26.2%), RA-like (4, 21.1%), PsA-like (4, 21.1%), and others (6, 31.6%). All RA-like patients had US findings indistinguishable from conventional RA patients. In addition, 3/5 (60%) of PMR-like patients had significant involvement of the hands and wrists. Abnormal findings on MRI or PET-CT were reported by clinical symptoms. No erosions or myofascitis were seen.”.

  1. Since only a few patients have MRI and PET I would not state that you were able to depict the main findings of these patients with these tools (discussion).

                 Thanks for pointing out this point. We have made some adjustments in the discussion.

We now state “There are few studies focused on the image findings in patients with irAEs induced by ICI. In the current work, we were able to describe several US, MRI, and 18F-FDG PET/CT findings in patients with ICI-induced arthritis and their association with the clinical pattern of presentation.”....” We included a relatively small cohort of patients with ICI-induced arthritis. The number of studies carried out, especially MRI and 18F-FDG PET/CT, allow a description of the most representative findings in patients with ICI-induced arthritis. Study of a larger number of patients to characterize these imaging findings better is required. However, so far, our sample is one of the largest series of patients with irAEs with imaging assessments.”

  1. Do you think that US is enough for the evaluation of these inflammatory joint disease? When MRI is needed? Please add a brief discussion on this with your experience, and citing also literature data.

Thanks for the comment. We believe that US is an affordable and safe technique that is very useful for an initial evaluation of patients with this inflammatory joint disease. In our opinion, MRI is reserved for unresolved issues, suspected early erosive disease, or muscle or myofascial involvement. We have added this in the discussion section. We now state “ In our patients, the US was the initial imaging test for evaluating the inflammatory joint disease in ICI-induced arthritis, due to easy accessibility. In our experience, the US re-solved clinical discrepancies in most cases. We proposed that the US could be used as the initial exploration,  and reserving MRI for unresolved issues, suspected early erosive disease, or muscle or myofascial involvement [24,34].

Reviewer 2 Report

The manuscript "Imaging Findings in Patients with Immune Checkpoint Inhibitors-induced arthritis" describes an important critical issue - ICIs-induced arthritis as ICIs have been used more often in cancer treatment recently.

Few comments on the manuscripts

1. Several typos and wrong word combinations in the texts. Please double-check or ask some professionals for English editing.

2. It is relatively unusual to mix an original study with a review paper in the same manuscript. The Discussion section mainly discusses the results from other groups, but the correlations between the current study and the references are not tight enough. 

3. The number of subjects is relatively small. Images obtained from cancer patients treated with ICIs but did not develop arthritis or from cancer patients with arthritis but without ICI treatment should be included and compared. It would help the readers the understand the whole picture of the study.

4. The conclusion is a bit weak. No critical findings or suggestions are proposed compared to other previous studies. 

Author Response

Reviewer #2

The manuscript "Imaging Findings in Patients with Immune Checkpoint Inhibitors-induced arthritis" describes an important critical issue - ICIs-induced arthritis as ICIs have been used more often in cancer treatment recently. Few comments on the manuscript:

  1. Several typos and wrong word combinations in the texts. Please double-check or ask some professionals for English editing.

 Thanks for the feedback. We have corrected the misspellings and grammatical errors. The text has been updated by a professional for English editing.

  1. It is relatively unusual to mix an original study with a review paper in the same manuscript. The Discussion section mainly discusses the results from other groups, but the correlations between the current study and the references are not tight enough.

 We appreciate you mentioning this point. The reasons for choosing this design correspond to the small number of patients included and that it is a pathology scarcely studied by imaging to date. We have tried to contrast our results with what has been published to date.

 We have added some changes to the discussion.

In our series of cases, US was performed after clinical assessment and as complementary information after physical examination. The US findings in patients with RA-like syndrome consist of marked synovitis/tenosynovitis involvement, although they were seronegative (anti-RF and ant-CCP), and presented high grades of SH and PD signal, with sonographic indistinguishable findings of a “classical RA”. These US findings do not differ from those found in other series of patients with synovitis/tenosynovitis induced by ICI and studied by US [23].

                It has been reported that ICI-induced PMR has US and FDG-PET/CT comparable to those seen in regular PMR [26]. We also found characteristic US findings in PMR-like ICI-induced patients similar to those previously described (i.e., glenohumeral and hip joint synovitis, bicipital tenosynovitis, and subacromial bursitis) [26]. Peripheral symmetrical synovitis and tenosynovitis, clinically subtle or not evident at all, in addition to rhizomelic involvement involvement was a notable finding. Although peripheral synovitis and tenosynovitis are frequently reported in PMR, it is usually oligoarticular and transient [31]. As in other studies of typical PMR [32], we found that in patients with ICI-induced PMR, tenosynovitis of the flexor tendons of the hand was the most frequent US finding of peripheral involvement. Interestingly, in treated patients, we found not synovitis, tenosynovitis, or bursitis in the shoulders and pelvic girdles, which seemed to have resolved with glucocorticoid treatment. However, peripheral synovitis/tenosynovitis in the hand joints remained evident.

  1. The number of subjects is relatively small. Images obtained from cancer patients treated with ICIs but did not develop arthritis or from cancer patients with arthritis but without ICI treatment should be included and compared. It would help the readers the understand the whole picture of the study.

                 Thanks for the comment. Our patients' evaluation with imaging tests was performed only on those who presented musculoskeletal symptoms. That is, they do not correspond to incidental findings in the tests requested within the follow-up of the underlying oncological pathology. Therefore, we do not have images that serve as a control. We have included a clarification in the section on the limitations of our study.

Certainly, our study has some limitations. We only conducted imaging tests in symptomatic patients. Therefore, the results do not include incidental findings in the requested imaging tests within the follow-up of the underlying oncological disease. As we mentioned above, sensitive techniques such as MRI or PET/CT scan can detect articular and periarticular subclinical inflammation in patients under ICI therapy. Therefore, articular symptoms could be underestimated on physical examination.

  1. The conclusion is a bit weak. No critical findings or suggestions are proposed compared to other previous studies.

Imaging techniques are helpful monitoring tools for diagnosing and following patients with ICI-induced arthritis. Since many patients have mild clinical symptoms or are under glucocorticoid treatment, a better definition of articular and periarticular structures is extremely useful in patient assessments. US findings in our study disclosed significant inflammatory changes, for instance, in patients with RA-like and PMR-like disease, as occur in chronic inflammatory arthropathies. These findings have implications for the diagnosis, treatment and prognosis, requiring, in some cases, DMARD therapies. Careful analysis of 18F-FDG PET, focusing on detecting signs of rheumatic irAEs could potentially help oncologists during follow-up to request a prompt assessment by rheumatologists. We propose that US could be used as the initial examination and reserve MRI for unresolved issues, suspected early erosive disease, or muscle or myofascial involvement. The wider of more sensitive techniques, such as MRI and PET/CT scan, will allow better recognition and definition of these novel and challenging rheumatic síndromes. 

Round 2

Reviewer 2 Report

Thanks for revising the manuscript. The content is significantly improved.